# Teaching Transformers Modular Arithmetic at Scale

## Abstract

Modular addition is, on its face, a simple operation: given $N$ elements in $\mathbb{Z}_q$, compute their sum modulo $q$. Yet, scalable machine learning solutions to this problem remain elusive: prior work trains ML models that sum $N \leq 6$ elements mod $q \leq 1000$. Promising applications of ML models for cryptanalysis—which often involve modular arithmetic with large $N$ and $q$—motivate reconsideration of this problem. This work proposes three changes to the modular addition model training pipeline: more diverse training data, an angular embedding, and a custom loss function. With these changes, we demonstrate success with our approach for $N = 256, q = 3329$, a case which is interesting for cryptographic applications, and a significant increase in $N$ and $q$ over prior work. These techniques also generalize to other modular arithmetic problems, motivating future work.

## 1 Introduction

Modular addition is an important operation commonly used in number theory and cryptography. The operation is simple: given $N$ elements $[x_1, x_2...x_N]$, $x_i \in \mathbb{Z}_q$, compute $s = \sum_{i=1}^{N} x_i \mod q$.

Despite its apparent simplicity, prior work has demonstrated that machine learning (ML) models struggle to perform modular arithmetic (Palamas, 2017; Lauter et al., 2024; Stevens et al., 2024). This is surprising because ML models can learn other complex math tasks such as symbolic regression, linear algebra, and computing the greatest common divisor (GCD) (Charton et al., 2021; Charton, 2022; 2024). Modular arithmetic, on its face, seems easier, but scalable ML solutions remain elusive.

Improved ML performance on modular addition could aid ongoing research efforts and open new research avenues. For example, modular arithmetic is a key component of many cryptographic hard problems, including Learning with Errors (LWE), which is the basis for post-quantum cryptosystems (PQC) standardized by NIST (Chen et al., 2022). ML models capable of modular arithmetic could aid nascent efforts to use ML models for cryptanalysis of LWE (Wenger et al., 2022; Li et al., 2023a;b; Stevens et al., 2024) or enable ML-powered cryptanalysis of other cryptosystems. Standardized LWE systems typically involve adding hundreds of random elements modulo $q$.

**Our Contribution.** Motivated by these potential use cases, we propose new methods enabling ML models to perform modular addition for a variety of $N$ and $q$, up to $N = 256$ and $q = 3329$. Our method significantly outperforms prior work, which summed $N \leq 6$ elements mod $q \leq 1000$, and generalizes to other modular arithmetic operations. In developing our methods, we first identify factors that limit models' ability to learn modular arithmetic: (1) lack of diverse training data, (2) lack of inductive bias for modular addition, and (3) unsuitable loss functions. We address these by:

- Constructing the **training data distribution** to ensure more diverse elements are represented.
- Introducing an **angular embedding** (inspired by Stevens et al. (2024)) that represents model inputs and outputs as coordinates on the unit circle, improving inductive bias for modular addition.
- Designing a **new loss function** with penalty term discouraging model convergence at local minima.

The remainder of this paper proceeds as follows. §2 discusses related work on ML-enabled modular arithmetic. §3 describes key limitations of prior work and our novel methods to overcome them. §4 presents key results on modular arithmetic problems with varying $N$ and $q$. §5 reports ablation studies over the methodology changes we introduce. §6 applies our methods to other asymmetric functions of interest, and §7 discusses future work.

| # Terms ($N$) | Mod ($q$) | MSE | % Accuracy | $\tau = 0.5\%$ Accuracy |
|---|---|---|---|---|
| 20 | 257 | $0.04 \cdot 10^{-4}$ | 99.9% | 100.0% |
| 20 | 769 | $0.03 \cdot 10^{-4}$ | 98.2% | 100.0% |
| 20 | 3329 | $0.04 \cdot 10^{-4}$ | 57.0% | 100.0% |
| 100 | 257 | $0.28 \cdot 10^{-4}$ | 97.8% | 99.9% |
| 100 | 769 | $0.32 \cdot 10^{-4}$ | 70.6% | 99.8% |
| 100 | 3329 | $0.42 \cdot 10^{-4}$ | 20.7% | 99.8% |
| 256 | 257 | $1.68 \cdot 10^{-4}$ | 95.8% | 99.8% |
| 256 | 769 | $0.63 \cdot 10^{-4}$ | 52.8% | 99.5% |
| 256 | 3329 | $0.46 \cdot 10^{-4}$ | 16.4% | 99.6% |

Table 1: **Our methods enable ML models to add $N \leq 256$ elements** mod $q \leq 3329$. All metrics are computed on a held out test set. MSE is mean squared error, % Accuracy is percentage of predictions exactly correct, $\tau = 0.5\%$ Accuracy is percentage of predictions within $0.005q$ of right answer (see §3 for details).

## 2 RELATED WORK

| Paper | # Terms ($N$) | Mod ($q$) | % Accuracy | Model Type |
|---|---|---|---|---|
| Nanda et al. (2023) | 2 | 53, 109, 113, 401 | 100 | Transformer |
| Mohamadi et al. (2024) | 2 | **433** | 100 | 2-layer MLP |
| Doshi et al. (2024) | **6** | 11, 23 | 97.1 | 2-layer MLP |
| Gromov (2023) | 2 | 97 | 100 | 2-layer MLP |
| Jelassi et al. (2023) | 2 | 100, **1000** | 73 | Encoder-only transformer |
| Abbe et al. (2024) | 2 | 2 | 100 | 4-layer MLP |

Table 2: **Summary of prior work on ML-enabled modular addition**. Best $N$ and $q$ are **bold**.

Prior work has investigated whether ML models can learn modular arithmetic operations (Palamas, 2017; Lauter et al., 2024; Gromov, 2023; Abbe et al., 2024; Mohamadi et al., 2024; Doshi et al., 2024). Table 2 summarizes the best prior results on modular addition specifically. The best existing methods train models that sum $N \leq 6$ elements for moduli up to $q = 1000$.

We scale ML-enabled modular addition to tackle larger $N$ and $q$, motivated by problems in number theory and cryptography that involve addition of many elements mod large primes. Prior work has laid groundwork for analytically understanding how models learn modular arithmetic (Gromov, 2023; Doshi et al., 2024). Our methods build on three observations from prior work:

- **Need for representative training data:** Mohamadi et al. (2024) showed that models need to be trained on a constant fraction of all possible modular arithmetic behaviors for a given $N$ and $q$ to generalize. This implies that better designed training datasets could aid learning.
- **Importance of appropriate model representations:** Nanda et al. (2023) showed that transformers trained to perform modular addition inherently learned to convert their inputs to polar coordinates, combine them, and then decode them back into the resulting integer sum. This suggests that models with inductive bias towards coordinate representations may perform better on this problem.
- **Importance of loss functions:** Several works attribute models' failure to learn more complex modular addition problems to the complexity of the loss space (Gromov, 2023; Jelassi et al., 2023). Because 0 and $q - 1$ are "close" in a modular field, seemingly different elements must map to the same loss region, making gradient descent difficult. A carefully designed loss function could help.

## 3 METHODOLOGY

Following prior work, we train models to add $N$ elements mod $q$ (fixed $N$ and $q$ for each model). Here, we list proposed improvements to the training pipeline that address the limitations described in §2. Then, we describe our end-to-end training procedure and evaluation metrics.

## 3.1 PROPOSED IMPROVEMENTS

**More Diverse Training Data to Improve Generalization.** Most prior work trains models using randomly generated $(\mathbf{x}, s)$ pairs, where $\mathbf{x}$ is drawn uniformly at random from $\mathbb{Z}_q^N$, i.e. $\mathbf{x}$ consists of elements $[x_1, x_2, \ldots, x_N]$, $x_i \in \mathbb{Z}_q$ and $s = \sum_{i=1}^N x_i \mod q$ (Jelassi et al., 2023; Doshi et al., 2024). Building on observations about the importance of training data diversity, we generate our training data using two distributions, $f$ and $g$. These help the model generalize by presenting it with simpler versions of the problem ($f$) and ensuring the full data distribution is well represented ($g$).

*Enabling gradual learning via $f$:* We postulate that models may learn better when they see "simpler" versions of the target operation, in this case modular sums with more zero elements. Seeing these simplified problems may help models understand the modular arithmetic structure and learn better. Thus, we propose adding additional *sparse* vectors to the training data, in which more coordinates of $\mathbf{x}$ are 0. To generate these, we fix a probability density function (PDF) $f : \{0, 1, \ldots, N\} \rightarrow [0, 1]$. Then, to create a training instance, we:

- Sample a random variable $z$, representing the number of zeros in each vector, from distribution $f$.
- Then, sample $N - z$ integers uniformly from the set $\{1, 2, \ldots, q - 1\}$. These integers, along with $z$ zeros, are used to construct a vector of length $N$.
- Lastly, shuffle the vector to ensure randomness in element order.

We experiment with three $f$s: $f_{\text{uni}}(z) = \frac{1}{N+1}$ (i.e. uniform density), $f_{\text{inv\_sqrt}}(z) \propto \frac{1}{\sqrt{z+1}}$ and $f_{\text{inv}}(z) \propto \frac{1}{z+1+\sqrt{N}}$, where $\propto$ means the functions are rescaled by a constant such that the sum of $f$ over all $z$ in its domain equals 1. We compare these to a baseline of $f_{\text{default}}$, which is the PDF of the number of zeros in $\mathbf{x}$ when $\mathbf{x}$ is drawn uniformly from $\mathbb{Z}_q^N$. Figure 1 shows the sparsity of examples created using these four sample strategies with $N = 20$ and $q = 257$.

*Representing distributional tails with $g$:* Wenger et al. (2024) observe that the sum of $N$ elements from $\mathbb{Z}_q$ follows the Irwin-Hall distribution, denoted as $g_{\text{default}}$ in Figure 2. Their analysis shows that for $N = 3$ the sum mostly falls in the range $[q, 2q]$, and models struggle predicting modulo $q$ sums of vectors $\mathbf{x}$ when their pre-modulo sum lies outside this range. To address this, we augment our training dataset with more instances whose sums (or equivalently their averages $\mu = \frac{1}{N}\sum_{i=1}^N x_i$ since $N$ is fixed) are in the distribution tails. In particular, we fix a new PDF $g : \{0, 1, \ldots, q - 1\} \rightarrow [0, 1]$ and to create training instances from $g$ we:

- Sample a random variable $\mu$, representing the target rounded average, from distribution $g$.
- Sample $N$ integers uniformly from the set $\{0, 1, \ldots, q - 1\}$. If the rounded average of these elements is exactly equal to $\mu$ we keep this sample, otherwise we repeat this step.

Again, we let $g_{\text{default}}$ be the PDF of $\mu(\mathbf{x})$ when $\mathbf{x}$ is drawn uniformly from $\mathbb{Z}_q^N$, pictured in Figure 2. Next, we introduce $g_{\text{interval}}$, also pictured in Figure 2, a uniform PDF over a centered range of $\mu$s and zero outside this range. Essentially, $g_{\text{interval}}$ is designed to overweight somewhat rare $\mu$ values from $g_{\text{default}}$, but to exclude very rare ones since we find these are very hard/expensive to generate. See Appendix A for a more formal definition.

*Dataset construction:* To create the training dataset, we sample repeatedly from $f$ and $g$ as defined above until we have sufficient training data. $f$ data can be generated easily, with no rejection sampling, but generating training samples from $g$ is computationally expensive since it requires significant rejection sampling. Because of this, we generate far fewer samples from $g$ than $f$. The exact ratio depends on $N$ and $q$, but $g$ samples typically compose less than 0.05% of the total dataset. During evaluation, we evaluate models on examples drawn uniformly at random from $\mathbb{Z}_q^N$.

**Inductive Bias via Transformer Model and Angular Embedding.** We address our observed lack of inductive bias towards modular arithmetic by adding an *angular embedding for input and output data* and using an *encoder-only transformer* model. Stevens et al. (2024) first introduced the angular embedding, which represents input integers mod $q$ as points on the unit circle. The intuition is that this better represents the structure of modular arithmetic, since on the unit circle 0 and $2\pi$—which corresponds to $q$—are close. Practically, the embedding encodes an integer $a \in \mathbb{Z}_q$ as an angle $\phi = 2\pi \frac{a}{q}$ and then as a point $(\cos(\phi), \sin(\phi)) \in \mathbb{R}^2$.

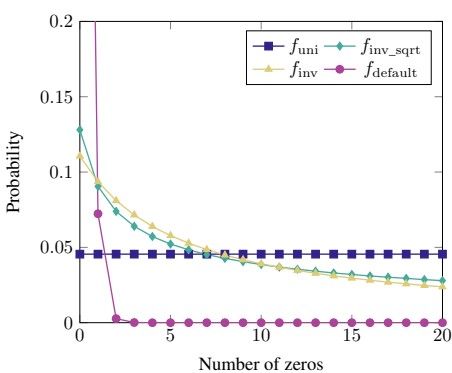

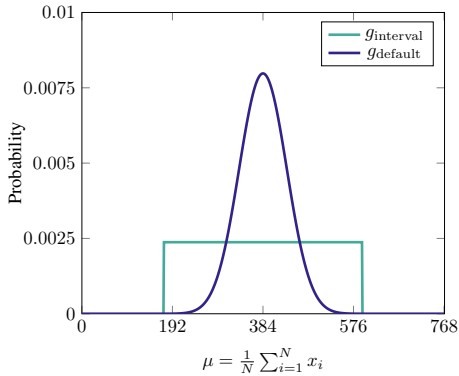

Figure 1: **Probability of number of zeros in each training data element when $N = 20$ and $q = 257$ for our three sampling distributions, $f_{\text{uni}}$, $f_{\text{inv}}$, and $f_{\text{inv\_sqrt}}$, and default sampling distribution $f_{\text{default}}$.** $f_{\text{inv}}$ and $f_{\text{inv\_sqrt}}$ produce more training data elements with $N$ nonzero elements compared to $f_{\text{uni}}$.

Figure 2: **Probability of $\rho(\mathbf{x})$ for each training data element when $N = 20$ and $q = 769$ for our two $g$ sampling distributions: $g_{\text{default}}$ and $g_{\text{interval}}$.**

Additionally, following Jelassi et al. (2023), we use an encoder-only transformer model, which provides two benefits. First, unlike MLPs used in much prior work on modular arithmetic, transformers have a self-attention mechanism that can capture relationships between the input elements and help the model learn to compute their sum. Second, an encoder-only transformer mirrors the structure of the problem, since modular addition involves an an input sequence but a single output token (Li et al., 2023b; Stevens et al., 2024; Jelassi et al., 2023), making it an appropriate choice.

**Custom Loss Function to Prevent Collapse.** Initially, we observed that the model would often converge on local minima like the origin of the unit circle, preventing the model from learning. To address this issue, we use a custom loss function during training that combines mean squared error (MSE) loss with an extra term. Given a prediction of the form $(x', y')$ and ground truth $(x = \cos\phi, y = \sin\phi)$, this loss takes the form:

$$\ell_\alpha = \alpha \left( x'^2 + y'^2 + \frac{1}{x'^2 + y'^2} \right) + (1 - \alpha) \left( (x - x')^2 + (y - y')^2 \right), \quad \alpha = 0.01$$

The first term penalizes the model for predicting the origin by driving the loss to infinity if $x' = 0, y' = 0$. It also encourages the model to predict $(x', y')$ on the unit circle (the first term is minimized with $x'^2 + y'^2 = 1$). The second term is the standard MSE loss. After some training $x'$ and $y'$ are close to the unit circle, therefore we can approximate $x'$ and $y'$ as $\cos\phi'$ and $\sin\phi'$. Under this condition, the MSE loss function component becomes:

$$\begin{aligned}
\ell &\approx (\cos\phi - \cos\phi')^2 + (\sin\phi - \sin\phi')^2 \\
&= \cos^2\phi - 2\cos\phi\cos\phi' + \cos^2\phi' + \sin^2\phi - 2\sin\phi\sin\phi' + \sin^2\phi' \\
&= 2 - 2\cos(\phi - \phi')
\end{aligned}$$

This loss component will be minimized when $\cos(\phi - \phi') \approx 1$, which occurs at $\phi - \phi' = 0$ and $\phi - \phi' = 2\pi$. In the modular arithmetic setting, we want 0 and $2\pi$ to be understood as "close" in the loss space, so this loss term correctly describes the desired behavior.

### 3.2 MODEL TRAINING AND EVALUATION

We implement the proposed changes and train models to sum $N$ elements mod $q$.

**Parameter Selection.** We experiment with $N = \{20, 50, 100, 256\}$ to identify trends as $N$ increases. Because we are interested in possible applications in cryptography, we use prime moduli, which are commonly used in that setting. We also tested with non-prime modulus $q = 1000$ and obtained similar results, as shown in Appendix D. We use $q = \{257, 769, 3329\}$, including one ($q = 3329$) used in a real-world cryptosystem, CRYSTALS-KYBER (Avanzi et al., 2021). We select $N = 20, q = 257$

as our base case for experiments because the sample space is large enough to ensure the model is generalizing.

**Training Procedure.** All our experiments were implemented in Python with Pytorch. We train the transformer models with a hidden dimension of 256, 4 attention heads, and 12 encoding layers on batches of 256 examples, using the Adam optimizer (Kingma & Ba, 2015) with a learning rate of $10^{-4}$, an initial linear warm-up phase of 1,000 optimization steps, and cosine scheduling. These parameters were chosen based on an extensive hyperparameter search (see Appendix B for more details). All experiments run on 8 V100 GPUs with 32 GB of memory. The models were trained for 30 epochs of 2.56 million examples per epoch per GPU. Training time is around 30 hours per GPU.

**Evaluation Metrics.** For evaluation, we generate a held-out test set $\mathcal{D}_{\text{test}}$ of size 100,000 that is distinct from the training set and contains examples drawn uniformly from $\mathbb{Z}_q^N$. To evaluate model performance on $\mathcal{D}_{\text{test}}$, we take the final hidden state of the transformer and pass it through a linear layer to produce an output of the form $(x', y')$. We project this point onto the unit circle, producing $(\cos \phi', \sin \phi') = (\cos \frac{2\pi}{q} s' \sin \frac{2\pi}{q} s')$ where $s' \approx s = \sum_{i=1}^{N} x_i \mod q$. The model prediction is then compared against the ground truth of $(\cos \frac{2\pi}{q} s, \sin \frac{2\pi}{q} s)$.

To get a complete picture of model performance, we compute the following metrics: Mean Squared Error (MSE) of angle predictions, % accuracy (correct/incorrect answer), and % accuracy with a margin of error ($\tau$) relative to $q$. MSE and % accuracy help us to evaluate the model's performance in terms of closeness between the predicted and ground truth angles (MSE) and predicted integer correctness (% accuracy). $\tau$-accuracy enables us to measure whether the model learns the approximate function behavior, even if exact accuracy is low. The formulae for these metrics are below:

$$\text{MSE} = \frac{1}{|\mathcal{D}|} \sum_{x \in \mathcal{D}} \left( (\cos \phi - \cos \phi')^2 + (\sin \phi - \sin \phi')^2 \right)$$

$$\text{Accuracy} = \frac{1}{|\mathcal{D}|} \sum_{x \in \mathcal{D}} \mathbb{1}_{s'=s}$$

$$\tau\text{-accuracy} = \frac{1}{|\mathcal{D}|} \sum_{x \in \mathcal{D}} \mathbb{1}_{\|s'-s\| \leq \tau q}$$

## 4 KEY RESULTS

Our methods enable models to learn modular addition of $N$ up to 256 elements mod $q$ up to 3329. We present best results across a range of $N$ and $q$ values in Table 3. All results are obtained from encoder-only transformer models with angular embeddings trained with the $f_{\text{inv\_sqrt}} + g_{\text{interval}}$ training data distribution and our custom loss function.

Overall, the MSE is near 0 across $N$ and $q$, showing that the model converges and learns well. Notably, $\tau = 0.5\%$ accuracy is near $100\%$ for all models. This means that in almost all cases, an "incorrect" model prediction is still within $0.5\%$ of $q$. For $q = 3329$, this means nearly all predictions are within $\pm 16$ of the correct answer. % accuracy declines as $N$ and $q$ increase. This decline is more notable when $q$ increases but $N$ is constant, suggesting that model performance is more tied to the magnitude of $q$ than that of $N$.

**Comparison to Prior Work.** We compare our results to a representative sample of prior work (Gromov, 2023; Doshi et al., 2024; Jelassi et al., 2023). Gromov (2023) and Doshi et al. (2024) train a multi-layer perceptron (MLP) and observe that the model learns modular addition via grokking (i.e. generalization occurs long after memorization), while Jelassi et al. (2023) use an encoder-only transformer similar to ours, but without our tweaks to data distribution, embedding, and loss function.

We implemented their approaches and trained models on $N = 20$, $q = 257$ data (our base case) with the same number of training data samples as we used. Table 4 reports results. We found that all three approaches had MSEs of 1.0 and % accuracies of less than 1%. In other words, the model does not learn the task at all. In comparison, our methods achieve 99.9% on the same problem.

Unlike Gromov (2023) and Doshi et al. (2024), we do not observe grokking in our models because we use a very small fraction of data from the possible sample space ($3.89 \cdot 10^{-40}$ when $N = 20$ and

| # Terms ($N$) | Mod ($q$) | MSE | % Accuracy | $\tau = 0.3\%$ Accuracy | $\tau = 0.5\%$ Accuracy |
|---|---|---|---|---|---|
| 20 | 257 | $0.04 \cdot 10^{-4}$ | 99.9% | 99.9% | 100.0% |
| 20 | 769 | $0.03 \cdot 10^{-4}$ | 98.2% | 100.0% | 100.0% |
| 20 | 3329 | $0.04 \cdot 10^{-4}$ | 57.0% | 99.9% | 100.0% |
| 50 | 257 | $0.13 \cdot 10^{-4}$ | 99.5% | 99.5% | 100.0% |
| 50 | 769 | $0.13 \cdot 10^{-4}$ | 88.5% | 99.8% | 100.0% |
| 50 | 3329 | $0.11 \cdot 10^{-4}$ | 35.2% | 99.8% | 100.0% |
| 100 | 257 | $0.28 \cdot 10^{-4}$ | 97.8% | 97.8% | 99.9% |
| 100 | 769 | $0.32 \cdot 10^{-4}$ | 70.6% | 99.4% | 99.8% |
| 100 | 3329 | $0.42 \cdot 10^{-4}$ | 20.7% | 99.1% | 99.8% |
| 256 | 257 | $1.68 \cdot 10^{-4}$ | 95.8% | 95.8% | 99.8% |
| 256 | 769 | $0.63 \cdot 10^{-4}$ | 52.8% | 98.2% | 99.5% |
| 256 | 3329 | $0.46 \cdot 10^{-4}$ | 16.4% | 98.5% | 99.6% |

Table 3: **Our methods perform consistently well adding** $N \in [20, 50, 100, 256]$ **elements** mod $q \in [257, 769, 3329]$. All metrics are computed on a held out test set. MSE is mean squared error, % Accuracy is percentage of predictions exactly correct, $\tau = 0.3\%$ Accuracy is percentage of predictions within $0.003q$ of right answer, and $\tau = 0.5\%$ Accuracy is percentage of predictions within $0.005q$ of right answer (see §3). The models perform with consistently low MSE and very high $\tau$-accuracies, but the exact accuracy declines with increasing $q$.

$q = 257$). As such, our models gradually learn with a standard training loss behavior and do not overfit.

| Method | MSE | % Accuracy | $\tau = 0.3\%$ Accuracy | $\tau = 0.5\%$ Accuracy |
|---|---|---|---|---|
| Gromov (2023) | 1.0 | 0.4% | 0.9% | 1.2% |
| Doshi et al. (2024) | 1.0 | 0.5% | 0.9% | 1.3% |
| Jelassi et al. (2023) | 1.0 | 0.3% | 0.7% | 0.9% |
| **Ours** | $0.04 \cdot 10^{-4}$ | 99.9% | 99.9% | 100.0% |

Table 4: **Our methods significantly outperform prior work for** $N = 20, q = 257$. We implemented the approaches described in previous work and evaluated all approaches with the same held out test set for $N = 20, q = 257$. MSE is mean squared error, % Accuracy is percentage of predictions exactly correct, $\tau = 0.3\%$ Accuracy is percentage of predictions within $0.003q$ of right answer, and $\tau = 0.5\%$ Accuracy is percentage of predictions within $0.005q$ of right answer (see §3 for details).

## 5 WHICH FACTORS MOST HELP MODELS LEARN MODULAR ARITHMETIC?

Next, we explore how our individual methods—diverse training data distribution, transformer model with angular embedding, and custom loss function—affect models' ability to learn modular arithmetic. Our goal is to understand performance gains provided by each relative to their combined effect.

### 5.1 EFFECT OF TRAINING DATA DISTRIBUTION

**Sparsity is Critical.** As described in §3, we construct more diverse training datasets by sampling elements defined by PDFs $f$ and $g$. Here, we explore how different sparsity PDFs ($f_{\text{default}}$, $f_{\text{inv}}$, $f_{\text{inv\_sqrt}}$, and $f_{\text{uni}}$, see §3) combined with $g_{\text{interval}}$ affect model performance. We report two metrics: % accuracy of models (exact accuracy) and the Kullback–Leibler (KL) divergence between the training and testing datasets. KL divergence quantifies the similarity between training dataset $\mathcal{D}_{\text{train}}$, constructed using functions $f$ and $g_{\text{interval}}$, and $\mathcal{D}_{\text{test}}$, sampled from the set $\mathbb{Z}_q^N$ uniformly at random, i.e. $f_{\text{default}}$. The results are in Table 5.

As Table 5 shows, the accuracy difference between models trained with the default sampling ($f_{\text{default}}$) and any other distribution $f$ is stark. The exact same architecture has 0% accuracy if we do not

modify the training dataset sparsity distribution and achieves over $85\%$ when we do. This strongly indicates that these models need to see sparse training examples to generalize.

| # Terms ($N$) | Mod ($q$) | Training Data $f$ | % Accuracy | KL divergence |
|---|---|---|---|---|
| 20 | 257 | $f_{\text{default}}$ | 0.4% | 0.0 |
| | | $f_{\text{inv}}$ | 99.6% | 29.9 |
| | | $f_{\text{inv\_sqrt}}$ | **99.9%** | 31.4 |
| | | $f_{\text{uni}}$ | 94.5% | 44.4 |
| 50 | 257 | $f_{\text{default}}$ | 0.4% | 0.0 |
| | | $f_{\text{inv}}$ | 97.2% | 69.9 |
| | | $f_{\text{inv\_sqrt}}$ | **99.5%** | 77.0 |
| | | $f_{\text{uni}}$ | 86.2% | 112.3 |
| 20 | 769 | $f_{\text{default}}$ | 0.1% | 0.0 |
| | | $f_{\text{inv}}$ | 93.5% | 37.8 |
| | | $f_{\text{inv\_sqrt}}$ | **98.2%** | 39.6 |
| | | $f_{\text{uni}}$ | 85.1% | 55.3 |

Table 5: **Sampling the training data from $f_{\text{inv\_sqrt}}$ produces the best accuracy results across $N$ and $q$.** % Accuracy is percentage of predictions exactly correct, KL divergence is the level of similarity between the training and testing datasets. With default sampling $f_{\text{default}}$, the model does not learn at all. Distributions with a KL divergence that is not too high or too low enable the model to perform best.

$\mathcal{D}_{\text{train}}/\mathcal{D}_{\text{test}}$ **KL Divergence Impacts Accuracy.** We observe that models trained on $f$ that produce very low ($\approx 0$) or very high $\mathcal{D}_{\text{train}}/\mathcal{D}_{\text{test}}$ KL divergence generalize worse than $f$ with mid-range KL divergence. Models trained with the default $f_{\text{default}}$ distribution have $0$ $\mathcal{D}_{\text{train}}/\mathcal{D}_{\text{test}}$ KL divergence, since the train/test distributions are almost identical, and model accuracy is $0\%$. On the other hand, the uniform sparsity function $f_{\text{uni}}$ diverges too far from the test distribution, resulting in lower accuracy. Distributions with fewer sparse training elements, like $f_{\text{inv}}$ and $f_{\text{inv\_sqrt}}$, perform best.

**Simple Examples Learned First.** Next, we validate our assumption that these models initially learn on simpler data (like sparse training examples) before learning the full task. To do this, we train a model on $N = 20$, $q = 257$ and monitor its performance on a dataset $\mathcal{D}_{\text{val}}$ drawn from the same distribution as $\mathcal{D}_{\text{train}}$. Figure 3 shows model accuracy on samples with 1 to 20 nonzero elements over 30 training epochs. Here, we see that the model initially performs better on sparse examples (e.g. 1 non-zero element) and then becomes accurate on more complex examples in later epochs. This suggests that these models first learn simpler sums and build on that knowledge to learn more complex sums, supporting our use of sparsity sampling in creating training data.

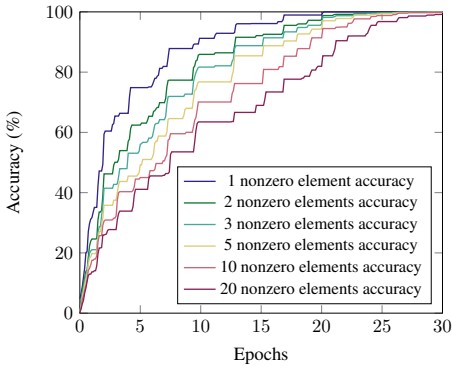

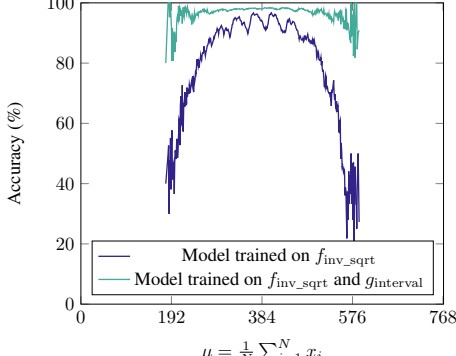

Figure 3: **The model learns to sum fewer nonzero elements earlier than more complex examples.** Model accuracy ($N = 20$, $q = 257$) after each epoch on unseen test set stratified by number of nonzero elements. As the number of nonzero elements increases, it takes longer for the accuracy to reach 100%.

Figure 4: **With training data from $g_{\text{interval}}$, the model is more consistently accurate on different $\mu$ ($N = 20$, $q = 769$).** Adding 0.02% of data from $g_{\text{interval}}$ significantly enhances the model's performance across both the tails and the central region of the $\mu(\mathbf{x})$ distribution.

**Importance of $g$ Data.** We also evaluate how the inclusion of training samples generated from $g$ affects model performance. As Table 6 shows, using training data from $f_{\text{inv\_sqrt}} + g_{\text{interval}}$ improves performance by $48\%$ on average compared to training data from $f_{\text{inv\_sqrt}}$ only. In contrast, training using data from $f_{\text{default}} + g_{\text{interval}}$ causes model performance to drop to 0%, emphasizing the need for sparse data. Figure 4 shows that models trained on $f_{\text{inv\_sqrt}}$ mainly struggle to classify elements in the tails of the $\mu$ distribution, but models trained on $f_{\text{inv\_sqrt}} + g_{\text{interval}}$ perform much better on these samples. Together, these results show that accuracy on the distributional tails can be greatly improved if a tiny amount (less than 0.05%) of data from distributional tails is added to the training set (similar to the priming technique from Jelassi et al. (2023)).

| # Terms ($N$) | Training data $f$ (Dataset size) | Training data $g$ (Dataset size) | % Accuracy Mod $q = 257$ | % Accuracy Mod $q = 769$ | % Accuracy Mod $q = 3329$ |
|---|---|---|---|---|---|
| 20 | $f_{\text{default}}$ (614.3M) | $g_{\text{interval}}$ (0.1M) | 0.4% | 0.1% | 0.0% |
|  | $f_{\text{inv\_sqrt}}$ (614.4M) | N/A | 99.5% | 93.0% | 31.7% |
|  | $f_{\text{inv\_sqrt}}$ (614.3M) | $g_{\text{default}}$ (0.1M) | 99.4% | 93.0% | 31.5% |
|  | $f_{\text{inv\_sqrt}}$ (614.3M) | $g_{\text{interval}}$ (0.1M) | **99.9%** | **98.2%** | **57.0%** |
| 50 | $f_{\text{default}}$ (614.2M) | $g_{\text{interval}}$ (0.2M) | 0.4% | 0.1% | 0.0% |
|  | $f_{\text{inv\_sqrt}}$ (614.4M) | N/A | 97.1% | 64.2% | 17.8% |
|  | $f_{\text{inv\_sqrt}}$ (614.2M) | $g_{\text{default}}$ (0.2M) | 96.9% | 64.3% | 17.2% |
|  | $f_{\text{inv\_sqrt}}$ (614.2M) | $g_{\text{interval}}$ (0.2M) | **99.5%** | **88.5%** | **35.2%** |
| 100 | $f_{\text{default}}$ (614.1M) | $g_{\text{interval}}$ (0.3M) | 0.4% | 0.1% | 0.0% |
|  | $f_{\text{inv\_sqrt}}$ (614.4M) | N/A | 89.7% | 39.0% | 9.2% |
|  | $f_{\text{inv\_sqrt}}$ (614.3M) | $g_{\text{default}}$ (0.3M) | 89.5% | 39.0% | 9.0% |
|  | $f_{\text{inv\_sqrt}}$ (614.1M) | $g_{\text{interval}}$ (0.3M) | **97.8%** | **70.6%** | **20.7%** |

Table 6: **Adding a tiny portion of data from a different distribution boosts the overall performances.** Test dataset is drawn uniformly from $\mathbb{Z}_q^N$. % Accuracy is percentage of predictions exactly correct. See §3.1 for definitions of $f$ and $g$.

**More Data Improves Performance.** Finally, we consider whether models can learn from fewer samples. We train models on $N = 20$, $q = 257$ with 1,000, 10,000, 100,000, 614.4M and 1,024M samples from the $f_{\text{inv\_sqrt}}$ sampling distribution only, with no data sampled from $g$. We arrive at the 614.4M and 1,024M cases because we generate data on the fly for each step and train for a fixed number of steps. In the other cases, we train the model over the fixed number of samples. As Table 7 shows, accuracy is highest in the 614.4M case, but results on limited data are encouraging. Even with $10{,}000$ samples, models can still sum elements with relatively high accuracy. We also see that having significantly more than 614M samples actually results in a decline in performance. We use the 614.4M samples for the rest of the experiments, unless otherwise noted.

| Dataset Size | $N = 20, q = 257$ % Accuracy | $N = 50, q = 257$ % Accuracy | $N = 20, q = 769$ % Accuracy |
|---|---|---|---|
| 1,000 | 39.2% | 6.5% | 16.6% |
| 10,000 | 96.7% | 90.5% | 73.5% |
| 100,000 | 99.2% | 95.0% | 91.0% |
| 614,400,000 (i.e. 614.4M) | **99.5%** | **97.1%** | **93.0%** |
| 1,024,000,000 (i.e. 1,024M) | 99.1% | 94.8% | 91.4% |

Table 7: **The model performs best when trained on 614.4M training examples.** We train the models with different numbers of examples (all with the $f_{\text{inv\_sqrt}}$ distribution and no $g$ distribution, angular embedding, and custom loss) and evaluate on the same test set for all. % Accuracy is percentage of predictions exactly correct.

## 5.2 EFFECT OF ANGULAR EMBEDDING

To understand the effect of the angular embedding on model performance, we evaluate models under four conditions: no angular embedding for input or outputs, input-only angular embedding, output-only angular embedding, and angular embedding for both input and output. When angular embeddings are not used for inputs, the model is trained on $N$-long integer sequences. When it is not used for outputs, the model predicts single integers. Experiments are run with varying $N$ and $q$.

As Table 8 shows, we achieve best results when the angular embedding is used for both the input and output. This table also shows that the output angular embedding more strongly impacts model accuracy than the input angular embedding. We hypothesize that this is because the model can learn the angular representation of inputs on its own, without the embedding (see Figure 7 in Appendix). In contrast, an integer (token) output overlooks the continuity from $q-1$ to 0, making it difficult for the model to learn the problem structure on its own. Using an angular embedding output of $(x, y)$, a position on the unit circle, implies that the output integer is in $\mathbb{Z}_q$ and makes learning easier.

| Embedding (Input) | Embedding (Output) | $N = 20, q = 257$ % Accuracy | $N = 50, q = 257$ % Accuracy | $N = 20, q = 769$ % Accuracy |
|---|---|---|---|---|
| Angular | Angular | **99.9%** | **99.5%** | **98.2%** |
| Integer | Angular | 99.6% | 97.6% | 92.8% |
| Angular | Integer | 82.5% | 72.3% | 56.3% |
| Integer | Integer | 73.2% | 9.1% | 0.5% |

Table 8: **Models perform better when trained with angular embeddings for both the inputs and outputs.** Models trained on the best settings identified in §5.1 and §5.3 and evaluated on the same test set for all. % Accuracy is percentage of predictions exactly correct.

## 5.3 EFFECT OF CUSTOM LOSS FUNCTION

Next, we consider the effect of our custom loss function on model performance. To do this, we train several models with varying $N$ and $q$ and two versions of the loss function given in §3.1: one with $\alpha = 0.01$, activating our additional term, and one with $\alpha = 0.0$, which is standard MSE loss. Table 9 reports our findings, averaged over 4 trials per setting.

| # Terms ($N$) | Mod ($q$) | (Best / Average / Worst) case Accuracy $\alpha = 0.01$ (Custom Loss) | $\alpha = 0.0$ (Standard MSE Loss) |
|---|---|---|---|
| 20 | 257 | **99.9%** / 99.8% / 99.8% | 96.5% / 77.2% / 53.8% |
| 50 | 257 | **99.5%** / 99.2% / 98.9% | 93.0% / 68.7% / 53.0% |
| 20 | 769 | **98.2%** / 97.9% / 97.3% | 84.2% / 73.5% / 65.8% |

Table 9: **Model consistently perform better when trained with our custom loss.** We train the models with the best training data parameters identified in §5.1 with angular embeddings and evaluate on the same test set for all. % Accuracy is percentage of predictions exactly correct.

Our custom loss function ($l_{\alpha=0.01}$) improves *best case* accuracy by 9% across all $N$, $q$ settings. Even more notably, it improves *average case* accuracy by 35%, compared to the standard MSE loss. The primary advantage of the custom loss is that it prevents model collapse, ensuring that the model consistently reaches high accuracy on every training run.

## 5.4 VISUALIZING LEARNED REPRESENTATIONS

Finally, we analyze the model's internal layers to understand how it represents output predictions. This helps us understand whether the model has conceptually "grasped" the problem. To do this, we pass input sequences to the model and extract their representations at different model layers. We perform Principal Component Analysis (PCA) with $k = 2$ components on the representation and plot them, coloring them based on the sum $s \bmod q$ of the input sequence. Figure 5 presents this analysis for three models trained with $q = 257$ and the following settings: $N = 10$ with the default $f_{\text{default}}$ training data distribution; $N = 20$ with the $f_{\text{default}}$ distribution; and $N = 20$ with the $f_{\text{inv\_sqrt}}$ distribution.

As Figure 5 shows, the $N = 10$ model with $f_{\text{default}}$ and $N = 20$ model with $f_{\text{inv\_sqrt}}$ both represent output predictions as points on a circle, indicating that they "understand" the problem. However, for the $N = 20$ setting without the custom distribution, the model fails to learn, and the representations are visually meaningless. This implies that for small $N$, the custom data distribution is not as important, likely because the problem is simpler, but for larger $N$, the custom distribution enables generalization.

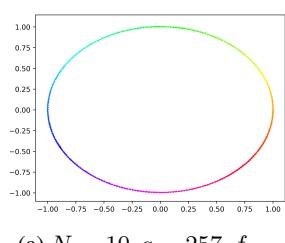 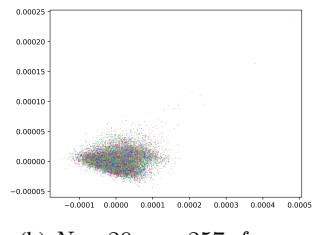 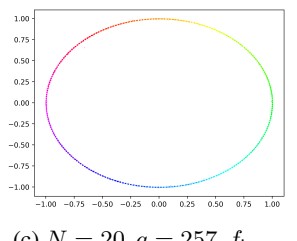

(a) $N = 10, q = 257, f_{\text{default}}$ (b) $N = 20, q = 257, f_{\text{default}}$ (c) $N = 20, q = 257, f_{\text{inv\_sqrt}}$

Figure 5: **Internal model representations for different $N$ and data distributions show that successful models learn the circular structure of the problem.** Plots show the first two PCA features for the model's internal representation after the output layer. Points with the same color have the same modular sum (i.e. they should be close together in representation). See Appendix C for more analysis.

## 6 BEYOND MODULAR ADDITION

Finally, we explore whether our methods enable ML models to learn other modular arithmetic functions beyond addition. Doshi et al. (2024) conjectured that two-layers MLPs can only learn functions that can be represented as $h(g_1(a_1), g_2(a_2), \ldots, g_N(a_N))$ and cannot extend beyond this class. We introduce a class of functions $h : \mathbb{Z}_q^N \to \mathbb{Z}_q$ outside the aforementioned class, where $h_{j,k} = \left( \sum_{i=1}^{N} a_i^j \right)^2 + a_1^k$, to show that our approach helps models learn other modular arithmetic functions. We train models to predict outputs from these functions, using the same setup as before: encoder-only transformer model with modified data distribution, input angular embedding, and custom loss. We also use a positional embedding in the transformer since these functions depend on input sequence positions.

Our results in Table 10 show that for $N = 20$ and $q = 257$, we achieve an accuracy exceeding **90%+** for these functions. This suggests that our methods can be applied to modular arithmetic in general, opening the door for further investigation.

| Function | % Accuracy |
|---|---|
| $h_{j=1,k=1} = (a_1 + a_2 + \ldots + a_N)^2 + a_1^1 \mod q$ | 90.3% |
| $h_{j=1,k=3} = (a_1 + a_2 + \ldots + a_N)^2 + a_1^3 \mod q$ | 91.0% |
| $h_{j=2,k=1} = (a_1^2 + a_2^2 + \ldots + a_N^2)^2 + a_1^1 \mod q$ | 90.5% |

Table 10: **With our methods, models can learn other modular arithmetic functions with good accuracy** ($N = 20, q = 257$). % Accuracy is percentage of predictions exactly correct.

## 7 DISCUSSION AND CONCLUSION

This work introduces novel techniques to help ML models learn modular addition. These techniques—varying the diversity of training data, using an angular embedding for model inputs and outputs, and introducing a regularized loss function—enable ML models to add hundreds of elements mod a large $q$ with high accuracy, a significant improvement over prior work. Our methods also enable models to learn other modular arithmetic functions, indicating their generalizability.

Several interesting directions remain for future work. First, as modulus size $q$ increases, our models have lower exact accuracy but consistently high $\tau = 0.5\%$ accuracy—above $99.5\%$. This motivates future work to understand this disconnect and improve performance as $q$ scales. Second, transferring our techniques to other settings (such as ML-enabled cryptanalysis) remains an open challenge. While our method achieves success on $q$ used in real cryptosystems and $N$ close to real-world use cases ($N = 512$ is used in practice (Avanzi et al., 2021)), transferring general modular addition knowledge to specific cryptanalysis tasks is nontrivial. Possible approaches include pretraining on this task and fine-tuning on specific application settings, but future research should consider creative approaches.

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

## A  Formal Definition of $g_{\text{interval}}$

Here, we provide a more formal definition of $g_{\text{interval}}(\mu)$, introduced in §3.1.

First, we fix $N$ and $q$. In order to define $g_{\text{interval}}(\mu)$, let $g_{\text{default}}(\mu)$ be the probability density function that follows from $\mu(\mathbf{x}) = \dfrac{1}{N} \sum_{i=1}^{N} x_i$ when $\mathbf{x}$ is drawn uniformly from $\mathbb{Z}_q^N$.

Let $\varepsilon$ be a real number. Given the $g_{\text{default}}$ distribution is centered at $\dfrac{q-1}{2}$, we build a symmetrical interval $I = \left[ \dfrac{q-1}{2} - k_\varepsilon, \dfrac{q-1}{2} + k_\varepsilon \right]$ where $k_\varepsilon$ is the smallest positive integer such that $\sum_{\rho \in I} g_{\text{default}}(\mu) \geq 1 - \varepsilon$.

We finally let

$$g_{\text{interval}}(\mu) = \begin{cases} \dfrac{1}{2k_\varepsilon + 1} & \text{if } \mu \in I \\ 0 & \text{if } \mu \notin I \end{cases}$$

We choose $\varepsilon = 10^{-5}$ for sampling reasons because generating samples for extremely rare $\mu$ takes too many rejection turns.

## B  Architecture Ablation

In §4, we report results using a transformer with 12 encoder layers and a hidden dimension of 256. We also train smaller models with 8 encoder layers and a hidden dimension of 256, as well as larger models with 12 encoder layers and a hidden dimension of 512. In Table 11, we report these results. Results are in line with those of §4. We select the architecture with 12 layers and a hidden dimension of 256 for all other experiments as it consistently produces high accuracy while training much faster than the model with 12 layers and a hidden dimension of 512.

|  |  | 8 layers 256 hidden dim | 12 layers 256 hidden dim | 12 layers 512 hidden dim |
|---|---|---|---|---|
| # Terms ($N$) | Mod ($q$) | % Accuracy | % Accuracy | % Accuracy |
| 20 | 257 | 99.7% | **99.9%** | 99.8% |
| 20 | 769 | 95.2% | **98.2%** | 97.3% |
| 50 | 257 | 94.0% | 99.5% | **99.6%** |
| 50 | 769 | 76.7% | 88.5% | **91.2%** |
| 100 | 257 | 79.5% | 97.8% | **98.1%** |
| 100 | 769 | 64.6% | **70.6%** | 65.2% |
| 256 | 257 | 78.2% | **95.8%** | 95.4% |
| 256 | 769 | 43.9% | 52.8% | **56.1%** |

Table 11: **Accuracy results for different transformer architectures across $N$ and $q$.** Results with $N \in [20, 50, 100, 256]$ elements mod $q \in [257, 769]$ for (a) smaller model, i.e. 8 layers and 256 hidden dimension, (b) chosen model, i.e. 12 layers and 256 hidden dimension and (c) larger model, i.e. 12 layers and 512 hidden dimension. % Accuracy is percentage of predictions exactly correct.

## C  Internal model representation

We show the output predictions as well as the internal model representations in Figure 6.

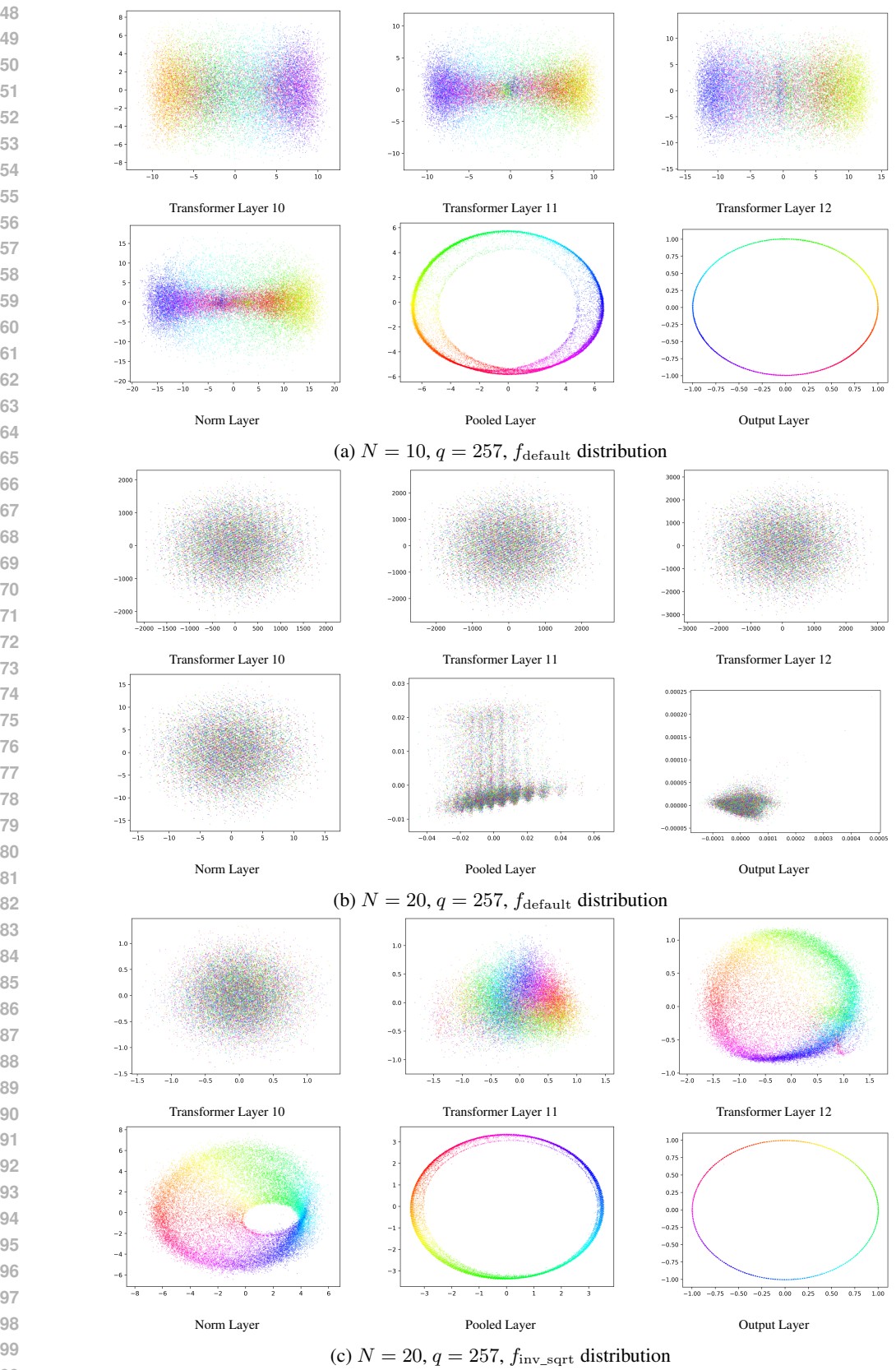

(a) $N = 10$, $q = 257$, $f_{\mathrm{default}}$ distribution

(b) $N = 20$, $q = 257$, $f_{\mathrm{default}}$ distribution

(c) $N = 20$, $q = 257$, $f_{\mathrm{inv\_sqrt}}$ distribution

Figure 6: **Internal model representations for different $N$ and data distributions show that successful models learn the circular structure of the problem.** Plots show the first two PCA features for the model's internal representation after each layer. Points with the same color have the same modular sum (i.e. they should be close together in representation).

# D ADDITIONAL RESULTS

We report additional results using our approach in Table 12 for $N \in [150, 384]$ and for a non-prime $q = 1000$. We see similar trends as Table 3.

| # Terms ($N$) | Mod ($q$) | MSE | % Accuracy | $\tau = 0.3\%$ Accuracy | $\tau = 0.5\%$ Accuracy |
|---|---|---|---|---|---|
| 20 | 1000 | $0.04 \cdot 10^{-4}$ | 95.1% | 100.0% | 100.0% |
| 50 | 1000 | $0.18 \cdot 10^{-4}$ | 80.1% | 99.7% | 99.9% |
| 100 | 1000 | $0.41 \cdot 10^{-4}$ | 57.2% | 99.4% | 99.8% |
| 256 | 1000 | $0.43 \cdot 10^{-4}$ | 50.8% | 99.3% | 99.8% |
| 150 | 257 | $1.21 \cdot 10^{-4}$ | 97.1% | 97.1% | 99.9% |
| 150 | 769 | $0.44 \cdot 10^{-4}$ | 65.8% | 99.3% | 99.8% |
| 150 | 1000 | $0.33 \cdot 10^{-4}$ | 57.4% | 99.4% | 99.9% |
| 150 | 3329 | $0.37 \cdot 10^{-4}$ | 18.5% | 98.9% | 99.8% |
| 384 | 257 | $2.80 \cdot 10^{-4}$ | 75.2% | 75.2% | 98.2% |
| 384 | 769 | $1.56 \cdot 10^{-4}$ | 35.2% | 94.2% | 97.7% |
| 384 | 1000 | $1.52 \cdot 10^{-4}$ | 33.7% | 94.6% | 97.5% |
| 384 | 3329 | $1.80 \cdot 10^{-4}$ | 8.6% | 90.7% | 98.1% |

Table 12: **Ablation for non-prime $q$ and $N \in [150, 384]$ elements.** All metrics are computed on a held out test set. MSE is mean squared error, % Accuracy is percentage of predictions exactly correct, $\tau = 0.3\%$ Accuracy is percentage of predictions within $0.003q$ of right answer, and $\tau = 0.5\%$ Accuracy is percentage of predictions within $0.005q$ of right answer (see §3 for details).

# E ADDITIONAL PLOTS

We show in Figure 7 the angular and the token (integer) PCA embedding representations with $N = 20$ and $q = 257$. These plots suggest that even without the angular embedding, the model is somewhat able to learn the circular representation on its own.

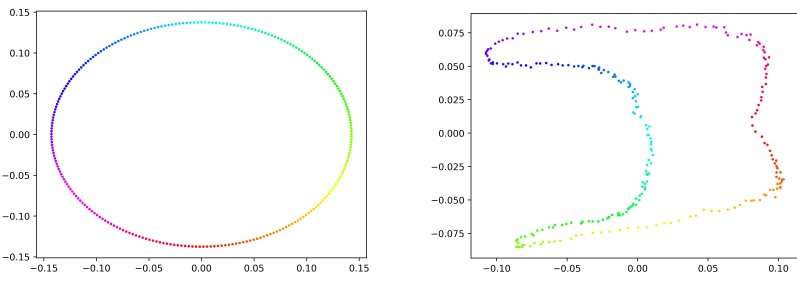

Figure 7: **PCA decomposition on angular (left) and token (right) embedding on trained model with $N = 20$ and $q = 257$**

