# OpenReview forum: "Teaching Transformers Modular Arithmetic at Scale"
_ICLR.cc/2025/Conference — ICLR 2025 Conference Withdrawn Submission_

### Official Review · Reviewer_WfY5 · 2024-10-25

**Soundness:** 4
**Presentation:** 3
**Contribution:** 2
**Rating:** 5
**Confidence:** 3

**Summary:**

The paper tackles the challenge of enabling machine learning models, specifically transformers, to handle modular arithmetic with significantly larger values for \( N \) and \( q \) than previously studied. Traditional ML models struggle with modular arithmetic, particularly with parameters like \( N = 6 \) and \( q = 1000 \). This work proposes three key modifications that together enhance the performance of transformers on modular addition tasks:

1. **Enhanced Training Data Diversity**: By including a mix of simpler and rare modular addition examples, the authors aim to help the model generalize effectively.
2. **Angular Embedding**: This technique maps integers onto a unit circle, aligning better with the periodic nature of modular arithmetic.
3. **Custom Loss Function**: The authors introduce a specialized loss function designed to prevent convergence on local minima, ensuring that the model learns effectively.

These methods enable the transformer-based model to achieve high accuracy on modular addition tasks with values up to \( N = 256 \) and \( q = 3329 \), significantly surpassing prior results. The approach also shows potential for generalization across other modular arithmetic functions.

**Strengths:**

This paper’s key strengths lie in its innovative methodology and rigorous validation. The angular embedding and specialized loss function introduce solutions directly tailored to the demands of ML-based modular arithmetic. As modular arithmetic is foundational in cryptography, this work could help drive advancements in ML-powered cryptanalysis. The methodological rigor is enhanced by detailed ablation studies, and the inclusion of visualizations like PCA plots adds clarity, reinforcing the paper's accessibility and value.

**Weaknesses:**

While the proposed data distribution and loss modifications are effective, they add complexity. Discussing potential simplifications or alternative approaches for less resource-intensive implementation would be beneficial. While the model performs well up to \( N = 256 \) and \( q = 3329 \), addressing potential limitations as these parameters increase would add further depth.

**Questions:**

1. **Local Minima at the Origin**: You mention that the model can converge on local minima like the origin of the unit circle, which hinders learning. Since the correct output for a label \( x \) would be represented as \( \cos(2\pi x / q) \) and \( \sin(2\pi x / q) \), could you clarify why the origin (0,0) acts as a local minimum in this context? It would be helpful to understand how this specific point prevents effective training, given the angular nature of the embeddings.

2. **Digit-wise Tokenization for Modular Addition**: Have you experimented with digit-wise tokenization methods, such as representing numbers as sequences of digits, to evaluate how the model performs on modular addition tasks? It could provide insights into the model's ability to generalize on addition when individual digits are tokenized.

3. **Comparison with Interpretability-focused Work**: In Table 2, many of the related works primarily address interpretability aspects rather than modeling improvements for modular addition. This focus makes direct comparison potentially less relevant. Could you elaborate on why these specific interpretability-focused works were chosen, and consider whether it might be beneficial to compare primarily with approaches that directly aim to enhance modular addition capabilities?

4. **Comparison with Other Embedding Techniques**: Given that you propose a new embedding and custom loss, it would be helpful to see how it compares with existing methods designed for modular arithmetic or general embedding approaches, such as abacus embedding (https://arxiv.org/abs/2405.17399) or dice embedding (https://aclanthology.org/2020.emnlp-main.384.pdf). Have you tried these methods, and if so, how did they perform relative to your angular embedding? This comparison could add further depth to your evaluation of embedding strategies in modular arithmetic tasks.

---

> ### Author Response · Authors · 2024-11-14
> **Response to Reviewer WfY5**
>
> We thank you for your thoughtful feedback and questions. We hope to address some of the weaknesses you mention in our future work (such as improving data efficiency and performance as $N$ and $q$ increase).
>
> To address your questions:
> 1. You are correct that the origin is not a local minimum in the classical sense. We observed that when trained with the standard MSE loss function, the model made predictions close to $(0, 0)$ for all inputs. This is likely due to the fact that the mean squared error loss function is minimized when the predicted values are close to the average value of the label. In this case, the label is represented as $\cos(2\pi x / q)$ and $\sin(2\pi x / q)$ as you said. Since the cosine and sine functions have a range of $[-1, 1]$ and are symmetric around 0, the average value of the target variable is close to 0. Therefore, the model is simply minimizing the loss function by predicting the average value of the target variable, which happens to be close to $(0, 0)$. See also Figure 6b in the Appendix to understand why the model predicts a constant value for all. We are happy to clarify this point in the revised version.
> 2. This is an interesting idea, thank you for the suggestion! We haven’t yet explored this but leave it for future work.
> 3. We conducted a comprehensive literature search and compared our approach to all the related work we can find. As far as we know, there aren’t any other approaches that directly aim to enhance modular addition capabilities, primarily because the interpretability works suggested that it may be a solved problem. However, our work shows that the existing methods are not sufficient for generalized modular arithmetic for larger N and q, thus motivating our approach.
> 4. Thank you for the suggestion to compare our approach to existing embedding methods like abacus and dice, it’s a great idea! We haven’t yet explored this but leave it for future work.

---

### Official Review · Reviewer_qJSi · 2024-11-02

**Soundness:** 3
**Presentation:** 3
**Contribution:** 2
**Rating:** 3
**Confidence:** 3

**Summary:**

The work considered learning modular addition via transformers at scale and proposed three changes to the modular addition model training pipeline for this purpose: 1) diversifying the training data; 2) an angular embedding, 3) a new loss function. The work showed that these changes lead to improvement for learning at scale, scaling up to N=256 elements modular q=3329. It also showed that these techniques generalize to other modular arithmetic problems (a few specific low degree polynomials modular q).

**Strengths:**

- The work investigated in detail the existing training methods on the problem, identified potential drawbacks, and proposed corresponding techniques to address them.
- The work provided empirical evidence that the proposed changes can help.

**Weaknesses:**

- The problem addressed is quite limited: scaling up for a specific problem of modular addition tested over uniform distirbution. While the work provided some motivation, it is still unclear what's the impact of the work for future research/applications.
- It is unclear if the technical contributions are significant. The changes proposed are natural and not surprising. Furthermore, although the work tested on a few other modular arithmetic problems, those problems are specific and the evaluation is quite preliminary. It is unclear if the techniques can help for more general learning settings eg other algebraic reasoning tasks.

**Questions:**

- What about using active learning/sampling to generate training data?
- The evaluation uses test data from a particular distribution (uniform). This is standard. But things can be different in applications. What if the test data (ie motivated via the cryptanalysis application mentioned in the intro) have a different distribution? How to adjust the techniques?

---

> ### Author Response · Authors · 2024-11-14
> **Response to Reviewer qJSi**
>
> We thank you for your feedback.  We’ve addressed your concerns about the importance of the modular addition task and the generalizability of the work in a “meta response” to all reviewers. We address the rest of your concerns below:
> * In hindsight the techniques we propose may seem “natural” but initially, even starting from a small number of summands like $N=10$ or $20$, models were not learning the modular addition task at all. A significant amount of experimentation on different curriculum strategies led to our current approach which we were eventually able to scale up to $N=256$.
> * We also show that the results are applicable beyond modular addition (especially more complex modular arithmetic functions) to show that the techniques are not specific to this one function.
>
> To answer your questions:
> * Using active learning is an interesting idea, and we would love to explore this in future work. Our current approach provides a straightforward and efficient way to generate training data for this problem.
> * In our current setup, the train and test distributions are different. We already show that we are able to achieve good performance when training on a modified distribution and testing on the uniform distribution. We also tried training and testing on the modified distribution (an easier task) and naturally achieved even better results. However, we didn’t include these results because this wouldn’t reflect a real world setting like cryptanalysis where the data is essentially random.

---

### Official Review · Reviewer_bxr9 · 2024-11-09

**Soundness:** 3
**Presentation:** 3
**Contribution:** 1
**Rating:** 3
**Confidence:** 4

**Summary:**

This paper designs an architecture, representation, and dataset to use to train an encoder-only transformer model to perform modular addition of a fixed number of addends modulo a fixed prime.

**Strengths:**

The techniques used do improve performance on this problem, sometimes drastically, and indeed escape the symmetry-based lower bounds of Mohamadi et al. (2024) by using non-uniform sampling and representations which are not permutation-equivariant. This aligns somewhat with the results of Abbe et al. (2024).

Some of the analyses of the impacts of different decisions in the training process are quite interesting.

**Weaknesses:**

The paper presupposes that it is interesting to train an ML model to perform modular arithmetic in order to get good performance. I would vehemently argue, despite the existence of several recent paper which do train ML models to perform modular arithmetic (many of which I do think are quite interesting and with whose details I am very familiar), that this is not of any interest whatsoever. Here is a function far more interesting to cryptanalysis for this task: `lambda q, nums: sum(nums) % q`. This function achieves 100% accuracy for any `N` and `q`, probably runs _many_ orders of magnitude faster than your trained model with _far_ less memory, and doesn't require 240 GPU-hours of training.

So why is there so much recent work on training ML models to do modular arithmetic? This is _because_ it's such an easy problem, where we can understand what the network is doing when, e.g., exhibiting grokking behavior, or thinking about curriculum design, etc. The focus of these papers is not on obtaining the best learned model, but on what the process of learning on this toy problem can tell us about learning in general.

Thus, a paper about obtaining the best ML model to do modular arithmetic seems entirely misguided to me. A paper using modular arithmetic as a case study to investigate problems like curriculum/training distribution design, out-of-distribution generalization, etc could potentially be very interesting! There are a few parts of this paper that touch on things along these lines, and indeed the decisions about representation, the training distribution you use, etc are intriguing. But they're in service of a useless problem. I would suggest instead taking the kinds of decisions you made here to get things to work as an idea to explore in more general cases, taking modular arithmetic as a test case, rather than trying to get the best modular arithmetic network.

**Questions:**

- Is there a cryptanalytic application where a transformer implementing modular arithmetic, or something close to it, would be preferable to simply calling highly-optimized and accurate modular arithmetic routines?

---

> ### Author Response · Authors · 2024-11-14
> **Response to Reviewer bxr9**
>
> We thank you for your feedback. We’ve addressed your concerns about the importance of the modular addition task and the generalizability of the work in a “meta response” to all reviewers. We address the rest of your concerns below:
>
> * Yes, the function you define could perform modular arithmetic faster and more effectively than a transformer.  However, as described in our responses to common reviewer concerns above, having techniques that help transformers learn modular arithmetic would aid ongoing work that uses ML models to attack hard problems in post-quantum cryptography. In this attack (see [1, 2, 3, 4] for details), transformers **learn a cryptographic secret as they learn modular arithmetic**. Having a model that can learn modular arithmetic thus isn’t a means to an end (otherwise a lambda function would be an acceptable substitute) but rather is the end in itself, when the problem is framed correctly with cryptographic inputs. Prior work suggests that these ML attacks would scale if better techniques (such as those in this work) were used to teach transformers modular arithmetic alongside the secret recovery task [4].
>
> * Regarding your question, the Learning with Errors problem (LWE) in cryptography described in the meta response is an example where having a transformer learn approximate modular arithmetic is valuable, and other approaches do not apply. In fact, the literature shows that the model does not have to be completely accurate on modular arithmetic or LWE to recover the secret vector [1, 2, 3].
>
> [1] Emily Wenger, Mingjie Chen, François Charton, and Kristin Lauter. [SALSA: Attacking Lattice Cryptography with Transformers.](https://proceedings.neurips.cc/paper_files/paper/2022/[file/e28b3369186459f57c94a9ec9137fac9-Paper-Conference.pdf) In Proc. of NeurIPS, 2022.
>
> [2] Cathy Yuanchen Li, Jana Sotáková, Emily Wenger, Mohamed Malhou, Evrard Garcelon, François Charton, and Kristin Lauter. 2023. [SalsaPicante: A Machine Learning Attack on LWE with Binary Secrets.](https://doi.org/10.1145/3576915.3623076) In Proceedings of the 2023 ACM SIGSAC Conference on Computer and Communications Security (CCS '23). Association for Computing Machinery, New York, NY, USA, 2606–2620.
>
> [3] Cathy Li, Emily Wenger, Zeyuan Allen-Zhu, François Charton, and Kristin Lauter. [SALSA VERDE: a machine learning attack on Learning With Errors with sparse small secrets.](https://proceedings.neurips.cc/paper_files/paper/2023/file/a75db7d2ee1e4bee8fb819979b0a6cad-Paper-Conference.pdf) In Proc. of NeurIPS, 2023.
>
> [4] Emily Wenger, Eshika Saxena, Mohamed Malhou, Ellie Thieu, and Kristin Lauter. [Benchmarking attacks on learning with errors.](https://eprint.iacr.org/2024/1229) In Proc. of IEEE Security&Privacy, 2025.

---

### Official Review · Reviewer_FM96 · 2024-11-09

**Soundness:** 3
**Presentation:** 3
**Contribution:** 2
**Rating:** 5
**Confidence:** 4

**Summary:**

This paper proposes a few techniques that promote faster convergence in learning modular addition with encoder-only transformers. The techniques include a slight modification of loss function, angular embedding of inputs and modifications of training distribution.

**Strengths:**

* The paper is mostly focused on experimental evaluation of different training strategies, and their experiments are well-detailed and reproducible.
* The methodology proposed is well presented and easy to understand and follow.

**Weaknesses:**

I've split my concerns into major and minor ones:

### Major concerns:
* I don't understand why solving modular addition in scale is important. The other papers that the authors have cited and compared their work to use the setting of modular addition as a means of studying different behaviours of training algorithms or models. The authors mention that it is important in cryptography literature, but they never elaborate on how "learning to solve modular addition" with the given inputs is an important task. If we have the angular embeddings, or the integers, or even one-hot embeddings, then solving the task is straightforward.

* **Same setting having different results:** In table 7, the numbers of the bold row (N=20, q=257) are different from the numbers in the first row of Table 8. Don't these represent the exam same setting in running experiments? If so, where is the discrepancy coming from? This setting appears in other tables with other (different) numbers as accuracy as well, which is confusing.

* Section 5.4: If I understand correctly, Figure 5 claims to depict the PCA visualization of the outputs. IIUC, the targets are the angular embeddings of modular sums, the output dimension is 2. I don't see why PCA is needed here, since the output dim is already 2. Furthermore, when MSE is low, it's clear that the outputs must correspond to the angular embeddings of the targets and must be distributed on a circle, and when MSE is high they should not. I don't see how this tells us anything about the internal workings of the model.

* Overall, I think the techniques proposed require a practitioner to know about the structure of the problem (that we're going to solve a modular addition problem) and are not general beyond modular arithmetic. On the contrary, when we know that we're dealing with a modular addition problem, there are far superior approaches to solve the task than learning a deep network.

### Minor concerns:
* IIUC, Mohamadi et al's claim regarding the need for a fraction of data only applies to the so called "kernel-regime" where the network is not allowed to do any feature learning, and doesn't apply to trained networks.
* For the cryptography use case that the authors have mentioned: does partial correctness (achieving non-trivial but also not 100% evaluation accuracy) matter in the mentioned use case? If not, how can one ensure 100% evaluation accuracy on a given task?

**Questions:**

I've mentioned my questions in the weaknesses section.

---

> ### Author Response · Authors · 2024-11-14
> **Response to Reviewer FM96**
>
> We thank you for your helpful feedback. We’ve addressed your concerns about the importance of the modular addition task and the generalizability of the work in a “meta response” to all reviewers. We address the rest of your concerns below:
>
> Major concerns:
> * **Same setting having different results**: We apologize for the confusion. In table 7 we provide the results for models trained only on the $f$ distribution, while everywhere else we provide results for models trained with $f + g$ distributions. We will edit and clarify this in the final version.
> * **Section 5.4**: In this section, we conduct PCA on the model’s internal representations after every layer. See Figure 6 for a more detailed version of the figure depicting the internal representations after each layer of the transformer. These results indicate that the circular representation starts to be learned in the hidden layers of the model (layer 12 in Figure 6c) when trained with our proposed changes as opposed to only in the pooled/output layers (Figure 6a). We will update this figure in a revised version to show the internal representations in layer 12 as opposed to the output layer.
>
> Minor concerns:
> * **Cryptography application**: Yes! The Learning with Errors problem (LWE) in cryptography described in the meta response is an example where approximate correctness of modular arithmetic is sufficient to achieve secret recovery. In fact, the literature shows that the model does not have to be completely accurate on modular arithmetic or LWE to recover the secret vector [1, 2, 3]
>
>
> [1] Emily Wenger, Mingjie Chen, François Charton, and Kristin Lauter. [SALSA: Attacking Lattice Cryptography with Transformers.](https://proceedings.neurips.cc/paper_files/paper/2022/[file/e28b3369186459f57c94a9ec9137fac9-Paper-Conference.pdf) In Proc. of NeurIPS, 2022.
>
> [2] Cathy Yuanchen Li, Jana Sotáková, Emily Wenger, Mohamed Malhou, Evrard Garcelon, François Charton, and Kristin Lauter. 2023. [SalsaPicante: A Machine Learning Attack on LWE with Binary Secrets.](https://doi.org/10.1145/3576915.3623076) In Proceedings of the 2023 ACM SIGSAC Conference on Computer and Communications Security (CCS '23). Association for Computing Machinery, New York, NY, USA, 2606–2620.
>
> [3] Cathy Li, Emily Wenger, Zeyuan Allen-Zhu, François Charton, and Kristin Lauter. [SALSA VERDE: a machine learning attack on Learning With Errors with sparse small secrets.](https://proceedings.neurips.cc/paper_files/paper/2023/file/a75db7d2ee1e4bee8fb819979b0a6cad-Paper-Conference.pdf) In Proc. of NeurIPS, 2023.

---

### Author Response · Authors · 2024-11-14
**Meta response to all reviewers**

We thank the reviewers for their input. We are glad the reviewers recognize that the paper “drastically” improves over prior work (Reviewer bxr9), presents “ innovative methodology and rigorous validation” (Reviewer WfY5), and is “well presented and easy to understand” (Reviewer FM96). Here, we address common concerns raised by reviewers. Responses to individual reviews are below the individual reviews.

**Concern 1: Importance of modular arithmetic (Reviewers FM96, bxr9, qJSi)**

Models that reliably learn and perform modular arithmetic would be valuable tools for cryptanalysis, in particular for post-quantum cryptosystems. For example, breaking the Learning with Errors (LWE) hard problem, upon which much post-quantum cryptography is built, requires reverse-engineering a subset sum in modular arithmetic (mod $q$). More formally, the LWE problem with binary secrets is: given **a**, an integer vector, and **b**, a noisy modular sum of certain elements of **a**, recover **s**, a vector representing which elements of **a** were summed.

Prior work has demonstrated that ML models can recover **s** when it is sparse, meaning only a few elements of a were summed [1, 2, 3]. Models struggle to scale to the denser **s** vectors used in practice (e.g. in standardized post-quantum cryptosystems like CRYSTALS-KYBER). Recent work [4, Section 6.3] indicated that the model’s ability to learn modular arithmetic limits attack scalability. The current secrets recovered are those for which **b** does not “wrap” around the modulus, indicating that if models better understood modular arithmetic, more complex secrets could be recovered. This motivates our work.

[1] Emily Wenger, Mingjie Chen, François Charton, and Kristin Lauter. [SALSA: Attacking Lattice Cryptography with Transformers.](https://proceedings.neurips.cc/paper_files/paper/2022/[file/e28b3369186459f57c94a9ec9137fac9-Paper-Conference.pdf) In Proc. of NeurIPS, 2022.

[2] Cathy Yuanchen Li, Jana Sotáková, Emily Wenger, Mohamed Malhou, Evrard Garcelon, François Charton, and Kristin Lauter. 2023. [SalsaPicante: A Machine Learning Attack on LWE with Binary Secrets.](https://doi.org/10.1145/3576915.3623076) In Proceedings of the 2023 ACM SIGSAC Conference on Computer and Communications Security (CCS '23). Association for Computing Machinery, New York, NY, USA, 2606–2620.

[3] Cathy Li, Emily Wenger, Zeyuan Allen-Zhu, François Charton, and Kristin Lauter. [SALSA VERDE: a machine learning attack on Learning With Errors with sparse small secrets.](https://proceedings.neurips.cc/paper_files/paper/2023/file/a75db7d2ee1e4bee8fb819979b0a6cad-Paper-Conference.pdf) In Proc. of NeurIPS, 2023.

[4] Emily Wenger, Eshika Saxena, Mohamed Malhou, Ellie Thieu, and Kristin Lauter. [Benchmarking attacks on learning with errors.](https://eprint.iacr.org/2024/1229) In Proc. of IEEE Security&Privacy, 2025.


**Concern 2: Generalizability of work (Reviewers FM96, bxr9, qJSi)**

A few reviewers noted that the scope of the work is too narrow or specific to modular arithmetic. Our findings on the importance of the data distributions and the “curriculum” may be generally applicable to other problems as well. For example, we show that the model learns “simpler” examples first even when we provide it with a shuffled dataset, suggesting that an explicitly defined gradual curriculum throughout training is not necessary. We also see the importance of providing varying complexities of the problem for model convergence. In addition, we show that adding a few examples from a different distribution ($g$) helps performance even though we evaluate on a completely different distribution than the training distribution. We also show that the results extend beyond modular addition (especially more complex modular arithmetic functions) to demonstrate that the techniques are not specific to only one function. However, we acknowledge that some of the proposed techniques (especially the custom loss and angular embedding) are specific to modular arithmetic problems.

As for superior approaches to “learning” modular arithmetic, of course there are algebraic ways to hard code modular arithmetic (e.g. based on the Euclidean algorithm), but then the model is not “learning” the task. As for learning the task, our paper is the current state-of-the art for summing many elements mod $q$ with a transformer as far as we know; feel free to provide any additional references we might have missed in our related work section.

---

> ### Comment · Reviewer_bxr9 · 2024-11-24
>
> Thanks for the extra details and references on LWE. The problem in LWE is not the problem of modularly summing several known numbers; instead, it is recovering $s$ from $(A, b)$ samples from the matrix equation $b = A s + e \pmod q$. While this is clearly a related problem, it is also certainly not the same problem, and a solution to your problem does not address the LWE problem. Some of the choices in representation and curriculum design that you've explored here seem like they could possibly apply to improving e.g. the SALSA attack [1] or related schemes. If that's the case, then you should try them on the interesting problem, not on a useless problem that's kind of sort of related to the interesting problem. I still don't see how the paper as submitted here is of interest to an ICLR audience, although it provides some foundations for you to do some future work that could be interesting.

---

> > ### Author Response · Authors · 2024-11-26
> >
> > Thank you for the feedback.

---

### Note · Authors · 2025-01-24

I have read and agree with the venue's withdrawal policy on behalf of myself and my co-authors.